# ACDC: ADAPTIVE CLOUD-DEVICE COLLABORATION FOR EFFICIENT AND ACCURATE SEMANTIC SEGMENTATION

## ABSTRACT

Semantic segmentation is a vital task in computer vision with wide-ranging practical applications. Advanced segmentation models achieve high accuracy but face computational challenges for device-based deployment, while lightweight models often produce coarse predictions, potentially losing pixel-level details. To address these limitations, this paper introduces the **Adaptive Cloud-Device Collaboration (ACDC)** framework, which combines the efficiency of device-side models with the robust capabilities of cloud-side models. ACDC employs an adaptive uncertainty detection mechanism to capture pixel-wise distributional shifts, filtering challenging samples for fine-grained processing on the cloud, and fuses dual-granularity predictions to achieve precise results. The framework comprises three key modules: Device-Aware Adaptive Segmentor (DAS) for coarse segmentation and uncertainty detection using a two-stage Uncertainty Decoupler; Dynamic Cloud Augmentation Module (DCAM) for processing challenging samples and adaptive update; and Collaborative Fusion Engine (CFE) for dual-granularity integration. Extensive experiments demonstrate that ACDC improves segmentation accuracy with minimal data transmission, adapts to dynamic environments, and effectively identifies uncertain samples. Code is provided in the supplementary materials.

## 1 INTRODUCTION

Semantic segmentationSzeliski (2022) is a fundamental task in computer vision with wide-ranging practical applicationsForsyth & Ponce (2002). Advanced deep neural networks, including models such as SegFormerXie et al. (2021), SETRZheng et al. (2021), and HRNetSun et al. (2019), have achieved significant progress in this domain. While these models deliver high performance, they are primarily suitable for cloud-based deployment due to their computational demands and large-scale architectures. In such setups, images are transmitted to the cloud for processing and segmentation, imposing substantial demands on cloud-device communication bandwidth and computational resources. This approach might introduce latency, thereby limiting its practicality for time-sensitive applications.

In this light, significant efforts have aimed at deploying segmentation models on resource-limited devices Howard (2017); Ma et al. (2018); Iandola (2016); Sandler et al. (2018). Existing optimization strategies fall into two categories: model lightweightingGuo et al. (2020); Han et al. (2015); Sun et al. (2023) and data lightweighting. *Model lightweighting* involves designing task-specific lightweight models to reduce computational cost. However, it might lack generalization to diverse data distributions. *Data lightweighting* is performed by reducing input resolution to enable faster inference but can lead to coarse-grained predictions due to downsampling. To address the problem of detail loss, some approaches refine these results by iterative upsamplingKirillov et al. (2020) or multi-stage enhancementZhang et al. (2021). However, upsampled pixels might incorporate inaccurate information, while multi-stage operations yield more precise results but introduce more computational overhead. This trade-off between refinement accuracy and computational efficiency poses challenges to their practical applicability.

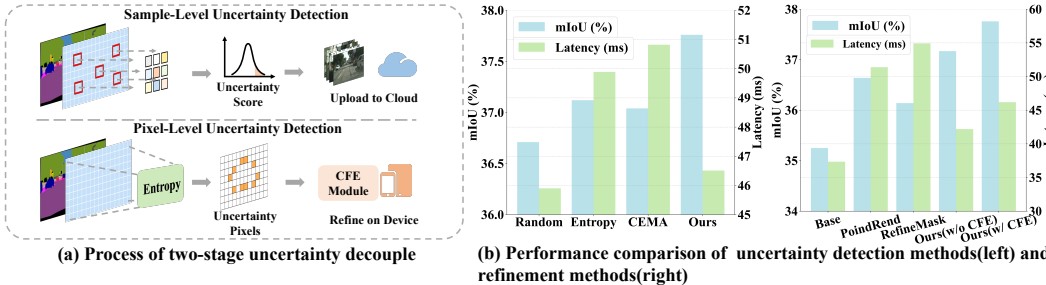

Figure 1: Illustration of the Uncertainty Decoupler and performance comparison of proposed methods. (a) illustrates the two-stage process of uncertainty decoupling. (b) compares the performance of different uncertainty detection and refinement methods.

In light of this, we propose to explore a cloud-device collaboration framework that unites the low-latency efficiency of on-device inference with the robust generalization of cloud models. Some general-purpose collaboration methodsKhani et al. (2021); Chen et al. (2024); Wang et al. (2022) have adopted similar principles, leveraging lightweight device models to handle easy samples while offloading challenging ones to the cloud for assistanceChinchali et al. (2021); Gan et al. (2023); Kang et al. (2017). However, their applicability to semantic segmentation remains constrained due to key limitations. Firstly, these methods are designed for classification tasks, relying on entropy-based uncertainty measures that are inadequate for the pixel-wise nature of segmentation. Small variations in pixel probability distributions can significantly affect uncertainty estimates, which traditional approaches fail to capture. Moreover, device-side models, constrained by computational resources, are more effective at capturing object contours from low-resolution outputs, which are crucial for delineating segment boundaries. In contrast, cloud-side models excel at extracting intricate internal object details from fine-grained outputs. Existing approaches fail to leverage both outputs simultaneously, instead relying exclusively on predictions from either the device or the cloud.

Existing methods provide trivial solutions to these limitations, yet they still face the following key challenges: 1) **Uncertainty Estimation**: A model may exhibit overconfidence in samples whose underlying distributions have shifted significantly. The challenge lies in comparing the pixel-wise output probability distributions with the model's capability and aggregating them to identify uncertainty based on distributional shifts. 2) **Prediction Fusion**: It is crucial to extract contour details from device-side predictions and object details from cloud-side predictions while devising a fusion strategy to integrate them with minimal computational overhead.

To address these challenges, we propose the **A**daptive **C**loud-**D**evice **C**ollaboration Framework (ACDC). The core idea is to employ an adaptive uncertainty detection mechanism that captures subtle pixel-wise distribution shifts, coupled with a dual-granularity fusion strategy that seamlessly integrates device-side contour information with cloud-side detailed outputs, thereby achieving superior segmentation accuracy with minimal computational overhead.

The framework includes three key components: **Device-Aware Adaptive Segmentor (DAS)**. This module deploys lightweight models that perform downsampling and coarse predictions to ensure low-latency processing. To differentiate uncertainty caused by model limitations and downsampling, we propose the Uncertainty Decoupler, which employs a two-stage mechanism as shown in Figure 1: *Sample-Level Uncertainty Detection*. Using a contrastive objective, we train an `Uncertainty Matrix` with pixel-wise distributions of positive and negative samples to represent the model's capability. During inference, distributional shifts are computed between sample probabilities and the matrix, assigning high uncertainty scores to samples with significant shifts, thereby addressing *Challenge 1*. *Pixel-Level Uncertainty Detection*. The image is upsampled to its original size, and pixel entropy is calculated to detect regions affected by downsampling inaccuracies. **Dynamic Cloud Augmentation Module (DCAM)**. On the cloud side, DCAM handles uncertainty samples and adaptively updates the device's uncertainty matrix. Using cloud outputs as pseudo-labels, it generates positive and negative samples to train a new matrix. To preserve learned parameters, DCAM performs weighted updates solely on parameters related to misclassified classes, minimizing disruptions to existing knowledge. **Collaborative Fusion Engine (CFE)**. To address *Challenge 2*, CFE employs a `Dual-granularity Fusion` mechanism for final output. First, CFE refines

the contour details from coarse predictions, focusing on high-entropy regions. Next, refined contour features are concatenated with cloud-side object details for further fusion, producing the final predictions.

Experimental results show that ACDC outperforms previous methods. We manage to enhance segmentation performance while minimizing upload volume, reducing upload costs. Regarding uncertainty detection, we validate that the proposed Uncertainty Decoupler identifies samples with greater distribution shifts, allowing cloud models to provide significant performance gains.

Our contributions are summarized as follows:

- We propose ACDC, a cloud-device collaboration framework which features a specialized uncertain samples detection method for segmentation and employs dual-granularity feature fusion to achieve precise results with low latency.
- We introduce Uncertainty Decoupler, modeling the capability distribution using `Uncertainty Matrix` to decoupling model uncertainty, and CFE module, extracting contour details and fine-grained outputs for complementary fusion with minimal computational overhead.
- Experimental results show that ACDC outperforms existing methods, showcasing adaptive capabilities for dynamic device environments and enhancing performance with reduced data transmission.

## 2 RELATED WORK

**Semantic Segmentation.** Semantic segmentation assigns class labels to image pixels. Early methods such as FCNLong et al. (2015) introduced pixel-wise prediction, while DeepLab variantsChen (2014); Chen et al. (2017; 2018) advanced the field using atrous convolutions and ASPP. PSP-NetZhao et al. (2017) added pyramid pooling for global context, and U-NetRonneberger et al. (2015) employed skip connections to recover fine details. More recent transformer-based architectures, including SegFormer and HRFormer, efficiently capture long-range dependencies and achieve high accuracy in domains like autonomous driving and medical imaging.

**Mask Refinement.** Mask refinement has long been used to improve segmentation quality Boykov et al. (2001); Felzenszwalb & Huttenlocher (2004); Shi & Malik (2000). PointRend Kirillov et al. (2020), drawing from graphics rendering, uses iterative subdivision to refine uncertain pixels at high resolution. RefineMask Zhang et al. (2021) embeds fine-grained features throughout a multi-stage pipeline for similarly precise outputs. In contrast, SparseRefine Liu et al. (2025) maintains essential details by applying sparse refinement to low-resolution predictions, significantly speeding inference without sacrificing accuracy.

**Cloud-Device Collaboration.** Cloud–device approaches exploit lightweight on-device models for effective inference and offload challenging cases to the cloudZhang et al. (2022); Fouladi et al. (2018); Crankshaw et al. (2017). Federated LearningMcMahan et al. (2017) preserves privacy via local updates but cannot adapt to real-time distribution shifts. Methods like DCCLYao et al. (2021) and AMSKhani et al. (2021) use cloud pretraining or knowledge distillation to improve on-device accuracy. However, they lack mechanisms to detect and prioritize the most challenging, pixel-level uncertainties, limiting their robustness under distributional shiftsVolpi et al. (2021); Marsden et al. (2022); Simon et al. (2022).

## 3 METHODOLOGY

Here we introduce our proposed ACDC. **Part of the theorem, proof, and methodology can be found in the Appendix due to the space limitation.**

### 3.1 PRELIMINARY

Consider a lightweight segmentation model $\mathcal{M}_d$ deployed on the device, where the input image is $\mathbf{x} \in \mathbb{R}^{H \times W \times C}$ and the output segmentation map is $\mathbf{y} \in \mathbb{R}^{H \times W}$. The segmentation task can be

Figure 2: Overview of the proposed Adaptive Cloud-Device Collaboration (ACDC) framework. Our ACDC comprises three main components: DAS and CFE on the device side, and DCAM on the cloud side. (a) illustrates the input samples. (b) details the architecture of the Device-Aware Adaptive Segmentor (DAS). (c) explains the Uncertainty Decoupler module. (d) presents the Dynamic Cloud Augmentation Module (DCAM). (e) showcases the Collaborative Fusion Engine (CFE) as the final stage. (f) outlines the adaptive update mechanism.

formulated as:

$$\mathbf{p} = \mathcal{M}_d(\mathbf{x}), \ \ \mathbf{y} = \arg\max_k \mathbf{p}_k, \ \forall k \in \{1, \dots, K\}, \tag{1}$$

where $\mathbf{p} \in \mathbb{R}^{H \times W \times K}$ represents the output probability distribution over $K$ classes.

In real-world applications, some input samples $\mathbf{x}_t \in \mathcal{X}_{\text{OOD}}$ may lie outside the training distribution, meaning they belong to an out-of-distribution (OOD) set $\mathcal{X}_{\text{OOD}}$ that $\mathcal{M}_d$ is unable to handle effectively. To address these OOD samples, we introduce a larger cloud-based model $\mathcal{M}_c$ that performs precise predictions on such samples:

$$\mathbf{y}_t = \arg\max_k \mathcal{M}_c(\mathbf{x}_t), \quad \text{where} \quad \mathbf{x}_t \in \mathcal{X}_{\text{OOD}}. \tag{2}$$

The overarching goal of our framework is to facilitate collaboration between the cloud and device models, leveraging their complementary strengths for high-quality segmentation while ensuring efficient inference.

$$\begin{cases} \underbrace{\mathcal{M}_d(\{\mathbf{x}_{(i)}\}_{i=1}^{|\mathcal{X}|})}_{\text{device model}} \xrightleftharpoons[\text{samples}]{\text{outputs, updates}} \underbrace{\mathcal{M}_c(\{\mathbf{x}_{(i)}\}_{i=1}^{|\mathcal{X}_{\text{OOD}}|})}_{\text{cloud model}} \\ \mathbf{y}_{\text{final}} = \text{Fusion}(\mathbf{y}_{\text{coarse}}, \mathbf{y}_{\text{fine}}) \end{cases} \tag{3}$$

## 3.2 ADAPTIVE CLOUD-DEVICE COLLABORATION

The Adaptive Cloud-Device Collaboration (ACDC) framework aims to obtain coarse-grained segmentation by device models with adaptive uncertainty detection and cloud-assisted fine-grained segmentation with low latency. The dual-granularity outputs are then fused to achieve precise results. As shown in Figure 2, on the device side, we propose the Device-Aware Adaptive Segmentor (DAS) and the Collaborative Fusion Engine (CFE) to achieve efficient, high-quality segmentation. The DAS incorporates an Uncertainty Decoupler to identify challenging samples, while the cloud server employs the Dynamic Cloud Augmentation Module (DCAM) to provide fine-grained segmentation for these samples.

### 3.2.1 DEVICE-AWARE ADAPTIVE SEGMENTOR (DAS)

Devices operating in real-world scenarios must contend with intrinsic variability and real-time requirements, necessitating continual adaption and low-latency inference. To tackle these challenges, we implement a lightweight model within the DAS module, which follows a two-step process. First, it downsamples the inputs and generates low-resolution predictions to achieve coarse-grained segmentation. Second, it leverages the Uncertainty Decoupler to effectively handle challenging samples, ensuring robust performance under dynamic conditions.

**Coarse segmentation.** Considering the latency requirements and computational constraints, achieving a balance between fine-grained segmentation accuracy and computational efficiency is challenging. To address this, we downsample the inputs to perform coarse segmentation. First, the resolution of the input $\mathbf{x}$ is halved to produce the coarse input $\mathbf{x}_c$. The device model $\mathcal{M}_d$ is then utilized to generate coarse predictions $\mathbf{y}_{\text{coarse}}$ as follows:

$$\mathbf{y}_{\text{coarse}} = \mathcal{M}_d \left( f_{\text{down}}(\mathbf{x}) \right), \quad \mathbf{y}_{\text{coarse}} \in \mathbb{R}^{\frac{H}{s} \times \frac{W}{s} \times C}. \tag{4}$$

Here, $f_{\text{down}}$ represents the downsampling operation, $\mathbf{x}_c$ is the downsampled input, and $\mathbf{y}_{\text{coarse}}$ denotes the corresponding coarse predictions while $s$ is the downsampling factor. Finally, the predictions are upsampled to match the original input resolution for subsequent processing.

**Uncertainty Decoupler.** To enhance generalization, our framework leverages the cloud server to process challenging samples beyond the device model's capabilities. We introduce the Uncertainty Decoupler to assess and filter these samples based on coarse predictions. The uncertainty in these predictions stems from two primary sources: (1) the device model's inherent limitations, and (2) the unavoidable information loss during downsampling. Traditional methods often fail to effectively decouple these distinct sources of uncertainty. To address this, we employ a two-stage mechanism for precise sample filtration.

*Sample-Level Uncertainty Detection.* The first stage is to identify samples that exceed the device model's capabilities. Traditional methods focus on classification tasks, neglecting the requirement of pixel-wise prediction in segmentation tasks. We propose training an uncertainty matrix $\mathcal{U} \in \mathbb{R}^{C \times C}$ to model the distribution of model capabilities and filter out samples with significant shifts based on pixel-level predictions. Each element $u_{ij}$ in $\mathcal{U}$ represents the uncertainty or distance between classes $i$ and $j$.

Let the model's predicted probability for an input $\mathbf{x}$ be $\mathbf{p}(\mathbf{x}) \in \mathbb{R}^{C \times \frac{H}{s} \times \frac{W}{s}}$. Then we define $\mathbf{q}(\mathbf{x}) \in \mathbb{R}^C$, where each element $\mathbf{q}_i(\mathbf{x})$ represents the probability that the input belongs to class $i$. We assume these probabilities follow a parameterized distribution family $\mathcal{F}(\theta)$, where $\theta$ are the parameters of the distribution. To enable the uncertainty matrix $\mathcal{U}$ to distinguish between positive and negative samples, we first compute the mean probabilities for positive ($\mathbf{q}_+$) and negative ($\mathbf{q}_-$) samples:

$$\mathbf{q}_+ = \mathbb{E}_{\mathbf{X}_+} \left[ \frac{1}{HW} \sum_{h=1}^{H} \sum_{w=1}^{W} \mathbf{p}(\mathbf{x}) \cdot \mathbb{1}_{\{\hat{y}=y\}} \right], \quad \mathbf{q}_- = \mathbb{E}_{\mathbf{X}_-} \left[ \frac{1}{HW} \sum_{h=1}^{H} \sum_{w=1}^{W} \mathbf{p}(\mathbf{x}) \cdot \mathbb{1}_{\{\hat{y}\neq y\}} \right]. \tag{5}$$

where $\mathbb{1}$ is an indicator function.

To effectively learn the uncertainty matrix $\mathcal{U}$, we define a contrastive loss function $\mathcal{L}(\mathcal{U}; \lambda, \gamma)$ that combines information from both positive and negative samples while incorporating a regularization term to prevent overfitting. The loss function is formulated as follows:

$$\mathcal{L}(\mathcal{U}; \lambda, \gamma) = (1 - \lambda) \cdot \mathbb{E}_{\mathbf{x}_+ \sim \mathcal{D}_+} \left[ \mathcal{S}_{\mathcal{U}}(\mathbf{x}_+) \right] - \lambda \cdot \mathbb{E}_{\mathbf{x}_- \sim \mathcal{D}_-} \left[ \mathcal{S}_{\mathcal{U}}(\mathbf{x}_-) \right] + \gamma \Omega(\mathcal{U}), \tag{6}$$

where $\lambda$ is the weighted factor, $\gamma > 0$ is the regularization strength, and $\Omega(\mathcal{U})$ is a regularization term, such as the Frobenius norm or trace of $\mathcal{U}$. $\mathcal{S}_{\mathcal{U}}(\mathbf{x})$ is the score function based on the uncertainty matrix $\mathcal{U}$, measuring the uncertainty of an input $\mathbf{x}$. Minimizing the contrastive loss function $\mathcal{L}(\mathcal{U}; \lambda, \gamma)$ ensures that the model can effectively distinguish between confident and uncertain predictions. The score function is defined as:

$$\mathcal{S}_{\mathcal{U}}(\mathbf{x}) = \mathcal{Q}(\mathbf{z}) \cdot \mathbf{q}(\mathbf{x})^{\top} \mathcal{U} \mathbf{q}(\mathbf{x}), \tag{7}$$

where $\mathcal{Q}(\mathbf{z})$ is a smooth function that captures the similarity between the uncertainty matrix $\mathcal{U}$ and the random matrix generated by the Gaussian kernel. Specifically,

$$\mathcal{Q}(\mathbf{z}) = \mathbb{E}_{\mathbf{z} \sim \mathcal{N}(0, \mathbf{I})} \left[ \exp \left( -\frac{1}{2} \| \mathcal{K}_{\sigma}(\mathbf{z}) - \mathcal{U} \|_F^2 \right) \right]. \tag{8}$$

Here, we introduce a Gaussian kernel function $\mathcal{K}_{\sigma}(\mathbf{z})$ to add smoothness and stability to the learning process, which is applied to the random variable $\mathbf{z}$, with $\| \cdot \|_F$ representing the Frobenius norm. The expectation $\mathbb{E}_{\mathbf{z} \sim \mathcal{N}(0, \mathbf{I})}$ ensures robustness against input variations. Using the uncertainty matrix $\mathcal{U}$, the uncertainty score $u(\mathbf{x}; \mathcal{U})$ for an input $\mathbf{x}$ is calculated as:

$$u(\mathbf{x}; \mathcal{U}) = \text{tr} \left( \mathbf{q}(\mathbf{x}) \mathcal{U} \mathbf{q}(\mathbf{x})^{\top} \right), \tag{9}$$

Table 1: Performance comparison of the proposed method and baselines on segmentation tasks. Results are averaged over five runs per dataset with varying random seeds, ensuring statistical significance at $p < 0.05$. The **best** performance is highlighted in bold and the second best is underlined.

| Datasets | ADE20K | | | Cityscapes | | | Pascal VOC | | | Pascal Context | | |
|---|---|---|---|---|---|---|---|---|---|---|---|---|
| Metric | mIoU | Latency(ms) | FLOPs(G) | mIoU | Latency | FLOPs(G) | mIoU | Latency | FLOPs(G) | mIoU | Latency | FLOPs(G) |
| Base | 35.25 | 37.35 | 95.48 | 65.30 | 40.67 | 188.00 | 63.47 | 40.24 | 93.96 | 39.34 | 39.76 | 82.98 |
| Tent | 35.15 | 95.95 | 95.80 | 63.78 | 87.61 | 195.12 | 63.76 | 85.46 | 97.38 | 39.12 | 88.36 | 83.20 |
| CoTTA | 35.82 | 84.73 | 148.07 | 66.97 | 81.34 | 288.01 | 64.20 | 81.57 | 146.28 | 40.70 | 85.64 | 130.31 |
| AMS | 36.58 | 51.83 | 118.03 | 66.85 | 54.70 | 239.33 | 64.75 | 54.26 | 116.35 | 40.75 | 52.31 | 102.58 |
| CEMA | 35.82 | 78.99 | 132.79 | 65.07 | 75.50 | 276.00 | 63.43 | 76.58 | 131.25 | 39.14 | 72.75 | 115.11 |
| Pseudo-label | 35.54 | 39.11 | 134.92 | 65.80 | 41.81 | 263.17 | 63.65 | 42.70 | 133.20 | 39.56 | 43.96 | 118.48 |
| CDCCA | 36.68 | 54.19 | 124.40 | 67.8 | 59.60 | 215.51 | 65.76 | 58.77 | 122.73 | 40.57 | 55.16 | 100.76 |
| Ours(5%) ($p < 0.05$) | 36.47 | 43.25 | 99.21 | 68.60 | 48.79 | 199.50 | 65.24 | 48.00 | 97.69 | 40.34 | 46.03 | 86.21 |
| **Ours(10%)** ($p < 0.05$) | **37.76** | 46.51 | 102.93 | **69.81** | 51.95 | 208.00 | **67.54** | 52.15 | 101.42 | **42.79** | 48.24 | 89.45 |

where $\text{tr}(\cdot)$ denotes the trace of a matrix. This score reflects the uncertainty of the input across different classes, with higher values indicating greater uncertainty. Finally, the device model filters the input images whose uncertainty score is larger than a threshold and uploads them to the cloud server.

$$\mathbf{x}_t = \{x \mid u_x > \tau_s\}. \tag{10}$$

*Pixel-Level Uncertainty Detection.* To address the information loss caused by the downsampling operation, we detect uncertain regions within the remaining samples by calculating the uncertainty for each pixel. we adopt entropy as the criterion for selecting these pixels. The decoupler uses the upsampled coarse predictions $\mathbf{p}' \in \mathbb{R}^{C \times H \times W}$ as input. The entropy for each pixel is then calculated as follow:

$$\mathcal{E}_{i,j} = -\sum_{k=1}^{C} \mathbf{p}'_{k,i,j} \log \mathbf{p}'_{k,i,j}. \tag{11}$$

Pixels with high entropy values are considered uncertain. The uncertain region mask $\mathbf{M}_{\text{uncertain}} \in \{0,1\}^{H \times W}$ is defined with a threshold $\tau$ as:

$$\mathbf{M}_{\text{uncertain},i,j} = \begin{cases} 1, & \text{if } \mathcal{E}_{i,j} > \tau, \\ 0, & \text{otherwise}. \end{cases} \tag{12}$$

These regions are subsequently processed by CFE module for further refinement.

### 3.2.2 Dynamic Cloud Augmentation Module (DCAM)

Upon receiving the challenging samples uploaded from the device, the cloud model leverages its powerful capabilities to generate fine-grained predictions, denoted as $\mathbf{y}_{\text{fine}}$. Additionally, to enable the device model to adaptively detect challenging samples in dynamic environments, the cloud model provides guidance by updating the device's uncertainty matrix.

**Decoupler adaptive update.** In continuously changing environments, the device model may struggle to accurately detect samples with varying distributions. To address this, the cloud model provides adaptive guidance to help the device model adjust to these distributions. The cloud model performs segmentation on the challenging samples and generates pseudo-labels to update the uncertainty matrix. To maintain the integrity of the existing learned parameters, a class-level mask is generated to identify the classes with discrepancies as follows:

$$\mathbf{M}_c = \mathbb{1}(\mathbf{y}_{\text{pseudo}} \neq \mathbf{y}_{\text{device}}), \tag{13}$$

where $\mathbb{1}$ is an indicator function. This selective updating mechanism ensures that only the incorrectly classified classes are modified, preventing disruption to the remaining parameters. Next, the new matrix $\mathcal{U}_{\text{new}}$ is calculated using the pseudo-labels. Finally, a weighted update is applied to the device model's uncertainty matrix, considering only the classes identified by the mask:

$$\mathcal{U}' = \left((1-\alpha) \cdot \mathcal{U} + \alpha \cdot \mathcal{U}_{\text{new}}\right) \cdot \mathbf{M}_c + \mathcal{U} \cdot (1 - \mathbf{M}_c), \tag{14}$$

where $\alpha \in [0,1]$ is the weight for the update. This final equation ensures that the uncertainty matrix is updated with a weighted combination of the old and new matrices, where the update affects only the classes identified by the mask. The cloud workflow is detailed in the Supplementary Material.

Table 2: Performance comparison of the proposed uncertainty detection mechanism and other methods on segmentation tasks. The **best** performance is highlighted in bold while the second best performance is underlined.

| Methods | ADE20K | | Cityscapes | | Pascal VOC | | Pascal Context | |
|---|---|---|---|---|---|---|---|---|
| | mIoU(%) | Latency(ms) | mIoU(%) | Latency(ms) | mIoU(%) | Latency(ms) | mIoU(%) | Latency(ms) |
| Random | 36.71 | 45.90 | 69.51 | 51.09 | 65.36 | 51.80 | 42.50 | 47.23 |
| Entropy | 37.12 | 49.89 | 69.75 | 52.22 | 66.38 | 52.65 | 42.62 | 48.83 |
| CEMA | 37.04 | 50.82 | 69.74 | 55.29 | 67.30 | 54.63 | 42.64 | 50.78 |
| **Ours** ($p < 0.05$) | **37.76** | 46.51 | **69.81** | 51.95 | **67.54** | 52.15 | **42.79** | 48.24 |

Table 3: Performance of the adaptive update mechanism with different filtration rate. The **best** performance is highlighted in bold.

| Filtration Rate | Datasets | ADE20K | Cityscapes | Pascal VOC | Pascal Context |
|---|---|---|---|---|---|
| 5% | w/o adaption | 36.30 | 68.29 | 65.02 | 40.26 |
| | w/ adaption | **36.47** *(+0.17)* | **68.60** *(+0.31)* | **65.24** *(+0.22)* | **40.34** *(+0.08)* |
| 10% | w/o adaption | 37.00 | 69.40 | 67.16 | 42.41 |
| | w/ adaption | **37.76** *(+0.76)* | **69.81** *(+0.41)* | **67.54** *(+0.38)* | **42.69** *(+0.28)* |
| 20% | w/o adaption | 39.01 | 70.66 | 69.14 | 43.75 |
| | w/ adaption | **39.32** *(+0.31)* | **71.43** *(+0.77)* | **70.04** *(+0.90)* | **43.92** *(+0.17)* |

### 3.2.3 COLLABORATIVE FUSION ENGINE (CFE)

To fully exploit the complementary strengths of coarse- and fine-grained predictions, we propose the Collaborative Fusion Engine (CFE), a single-stage module executed on the device. CFE first refines the coarse predictions—whose lower resolution nonetheless preserves essential contour information—using a modified Minkowski-UNet with sparse convolutions to effectively capture and enhance edge features. The resulting contour maps are concatenated with the cloud-generated fine-grained outputs and passed through lightweight convolutional layers that fuse these dual-granularity representations. By combining high-level semantic details with precise boundary information in one efficient step, CFE significantly improves overall segmentation accuracy while adding minimal computational overhead. The detailed workflow of device is shown in the Supplementary Material.

$$\mathbf{y}_{\text{aug-coarse}} = \mathcal{M}_u(f_{\text{up}}(\mathbf{y}_{\text{coarse}})) \quad \mathbf{y}_{\text{final}} = \text{Fusion}(\mathbf{y}_{\text{aug-coarse}}, \mathbf{y}_{\text{fine}}). \tag{15}$$

## 4 EXPERIMENTS

### 4.1 EXPERIMENTAL SETUP

**Datasets.** We evaluate ACDC on four pixel-annotated segmentation benchmarks: ADE20K (150 classes), Cityscapes (19 classes), Pascal VOC (21 classes), and Pascal Context (classes). To verify generality and efficiency, we perform ablations of each module on all datasets, using mean Intersection-over-Union (mIoU) as the primary metric.

**Baselines.** Our method is evaluated against multiple baselines, categorized into two groups: **Device-only** and **Cloud-Device Collaboration** approaches.

- **Device-only:** *Base* modelSun et al. (2019) evaluates the performance of a pre-trained device model on different datasets without any collaboration or adaption. *Tent*Wang et al. (2020) is s test-time adaption method that fine-tunes the batch normalization layers during inference to adapt the model to challenging samples.

- **Cloud-Device Collaboration:** *Pseudo-label*Lee et al. (2013) uses the labels generated by the cloud model to supervise and improve the device model. *CoTTA*Wang et al. (2022) is a continual test-time adaption approach that addresses error accumulation and catastrophic forgetting by leveraging the cloud model. *AMS*Khani et al. (2021) employs the cloud model to distill knowledge to the device model, overcoming resource limitations on the device. *CEMA*Chen et al. (2024) and *CDCCA*Wang et al. (2024) apply specialized detection methods on the device to select important samples for cloud-based distillation. The

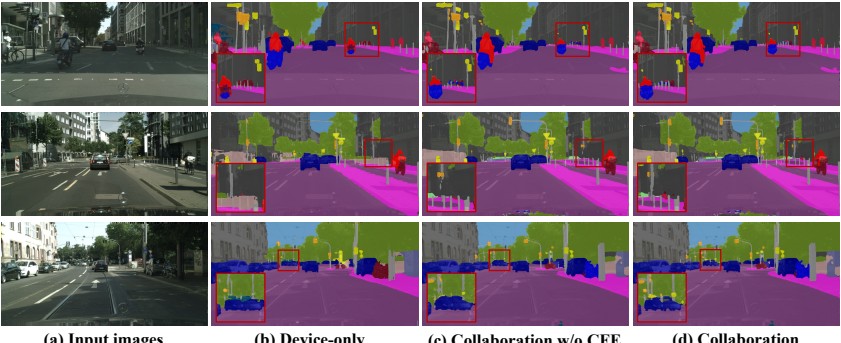

| (a) Input images | (b) Device-only | (c) Collaboration w/o CFE | (d) Collaboration |

Figure 3: Semantic segmentation results on Cityscapes. The collaboration method enhances the device model's segmentation quality, while the CFE module effectively integrates dual-granularity results for improved object details and contour precision.

former utilizes entropy and informativeness as detection criteria, while the latter employs a two-stage filtering process to determine which samples to upload.

**Implementation Details.** On the device, we employ HRNet-W48, which maintains high-resolution feature maps through parallel multi-scale branches, for coarse predictions. For the cloud, we use a transformer-based model, SegFormer-B5, to perform fine-grained predictions. The distinct architectures introduce diverse data distributions, enabling them to leverage complementary information from each other. Both models are initialized with their respective pre-trained weights on each dataset.

## 4.2 RESULTS AND ANALYSIS

Table 1 compares our method with other approaches across four datasets. The first two rows represent Device-only methods, while rows 3 to 7 correspond to Cloud-Device Collaboration baselines. Based on the results, the following conclusions can be drawn: (1) The *Tent* method performs worse than the *Base* method on most datasets except for Pascal VOC, indicating that directly updating device models based on test data is unsuitable for segmentation. (2) Among collaboration approaches, *CDCCA* leads, followed by *AMS*. Both methods demonstrate that selectively uploading challenging samples can improve overall segmentation accuracy. However, their uncertainty methods still fail to capture fine-grained, pixel-wise semantic shifts, limiting their performance. (3) Our proposed ACDC achieves notable performance improvements. At a 5% filtration rate, our method matches or exceeds the best baseline, while reducing inference latency by 17.16% on average. When the filtration rate increases to 10%, ACDC outperforms all baselines, achieving an average improvement of 3.17% in mIoU while reducing latency by 11.51% and FLOPs by 12.72% compared to the best baseline. These results confirm that ACDC strikes an optimal balance between generalization and efficiency. Its Uncertainty Decoupler reliably filters samples with large distributional shifts, and increasing the filtration rate identifies more challenging cases, leading to substantial mIoU improvements, particularly on difficult datasets such as ADE20K and Pascal Context.

In addition, we present qualitative results in Figure 3. As shown in the second column, the device model struggles to accurately classify distant pixels due to coarse inputs. The collaboration method without CFE improves classification with the cloud model but fails to capture some contours. Leveraging dual-granularity, our method (column **d**) achieves finer segmentation with accurate contour delineation. Further discussions are provided in the Appendix.

## 4.3 ABLATION STUDIES

### 4.3.1 EFFECT OF UNCERTAINTY DECOUPLER IN DAS

Table 2 highlights the comparison of model performance under different uncertain sample detection methods: *Random* method selects uncertain samples uniformly across the dataset, leading to the

poorest performance. This demonstrates its inability to identify meaningful uncertainties, especially in segmentation tasks. *CEMA*, which integrates entropy with informativeness, achieves results comparable to the simpler *Entropy* method. However, these approaches are constrained by their focus on image-level detection, disregarding the pixel-level semantic information essential for segmentation tasks. In contrast, our *Uncertainty Decoupler* addresses this limitation by employing an uncertainty matrix to aggregate pixel-level semantic information, thereby computing more precise uncertainty scores. This approach achieves the highest performance gains in terms of mIoU, significantly outperforming other methods, as shown in Figure 1(b). These results underscore the superiority of leveraging pixel-level cues over traditional image-level entropy for uncertainty estimation in segmentation, validating it as a more accurate mechanism of identifying the most uncertain samples.

### 4.3.2 EFFECT OF ADAPTIVE UPDATE IN DCAM

To assess DCAM's adaptive update mechanism, we performed an ablation study at various filtration rates (Table 3). In every case, adaptive updates outperformed the static baseline, confirming that dynamic parameter tuning enhances the detection of difficult samples. On the Cityscapes and Pascal VOC, mIoU gains increased steadily, reaching their maximum at a 20% filtration rate. For the other two datasets, performance improvements plateaued beyond a 10% filtration rate, indicating that moderate filtration rates focus on truly challenging inputs, while higher rates begin to include more samples already manageable by the device, which contribute little. Further discussion about filtration rate can be found in the Appendix.

### 4.3.3 EFFECT OF COLLABORATIVE FUSION ENGINE (CFE)

To quantify the benefits of our CFE module, we first conducted an ablation study focusing on its core U-Net component. As detailed in Table 4, comparing our approach with a standard U-Net and the point-cloud based model DGCNN, we find that the point-cloud ones outperformed because edge features in coarse segmentation resemble sparse point-cloud data, while our Minkowski-UNet achieved the best, demonstrating its superior ability to capture sparse contour information in 2D. Next, we evaluated various refinement strategies before final output on the ADE20K (see Table 5). While these methods improved segmentation accuracy, they introduced substantial computational latency. In contrast, CFE achieves superior performance with lower latency by effectively fusing dual-granularity segmentations: coarse predictions establish global structure and guide contour refinement, while fine-grained outputs supply rich semantic details. This integration of complementary strengths enhances overall segmentation. Additional ablations on module combinations and distance metrics are detailed in the Appendix.

Table 4: Performance comparison of our U-Net in CFE and other U-Net structures, *i.e.,* standard U-Net and DGCNN.

| Methods | Datasets | | | |
|---|---|---|---|---|
| | ADE20K | Cityscapes | Pascal VOC | Pascal Context |
| U-Net | 36.31 | 68.78 | 65.19 | 41.31 |
| DGCNN | 37.09 | 69.35 | 66.75 | 41.75 |
| **Ours** | **37.76** | **69.81** | **67.54** | **42.79** |

Table 5: Performance comparison of our CFE and other refinement methods on ADE20K.

| Methods | Metric | |
|---|---|---|
| | mIoU(%) | Latency(ms) |
| Base | 35.25 | 37.35 |
| +PointRend | 36.64 | 51.39 |
| +RefineMask | 36.14 | 54.92 |
| +Ours(w/o CFE) | 37.17 | 42.19 |
| **+Ours(w/ CFE)** | **37.76** | 46.51 |

## 5 CONCLUSION

We introduce ACDC, a cloud–device collaboration framework that delivers precise semantic segmentation by fusing dual-granularity predictions with an adaptive Uncertainty Decoupler. By separating model-capacity and downsampling uncertainties via an `Uncertainty Matrix`, ACDC minimizes data uploads and outperforms baseline methods. Its selective offloading of challenging samples and streamlined fusion of coarse and fine outputs make it well suited for resource-constrained applications, such as autonomous driving, medical imaging, and VR/AR, where high accuracy and low latency are essential.

## ETHICS STATEMENT

This work adheres to the ICLR Code of Ethics. We present ACDC, a cloud–device collaboration framework designed to improve semantic segmentation in resource-constrained settings, with applications in autonomous driving, medical imaging, and VR/AR. All datasets used in this study are publicly available and widely recognized in the vision community, ensuring compliance with licensing and usage guidelines. No personally identifiable information was involved, and no human or animal subjects were part of this research.

ACDC is designed to minimize unnecessary data transmission by selectively offloading only uncertain samples, thereby reducing potential privacy risks and promoting efficient deployment. We are committed to ensuring that this research is used responsibly and benefits domains where accuracy, safety, transparency, and fairness are paramount.

## REPRODUCIBILITY STATEMENT

To ensure the reproducibility of our work, we will open-source the implementation of the ACDC method. The complete code, along with detailed documentation, is included in the supplementary materials submitted with this paper. Additionally, upon acceptance, the code will be made publicly available on GitHub to facilitate further research and application.

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

## A APPENDIX

### A.1 PSEUDO CODE

---

**Algorithm 1** Segmentation Process on Device

---

**Input:** input image $\mathbf{x}$, device model $\mathcal{M}_d$, uncertainty matrix $\mathcal{U}$, upload threshold $\tau_s$, , downsample function $f_{\text{down}}$, upsample function $f_{\text{up}}$, entropy-based function $\mathbf{Ent}(\cdot)$, entropy mask $\mathbf{M}$, U-Net model $\mathcal{M}_u$, fusion function $\text{Fusion}(\cdot, \cdot)$

**Output:** entropy uncertain mask $\mathbf{M}$, coarse prediction $\mathbf{y}_{\text{coarse}}$, final output $\mathbf{y}_{\text{final}}$

  — **Device-Aware Adaptive Segmentor** —

 1: Calculate coarse prediction: $\mathbf{y}_{\text{coarse}} = \mathcal{M}_d(f_{\text{down}}(\mathbf{x}))$

 2: Estimate sample-level uncertainty: $u = \text{tr}(\mathbf{q}(\mathbf{x})\mathcal{U}\mathbf{q}(\mathbf{x})^\top)$
                     ▷ *Uncertainty decouple stage 1*

 3: **if** $u > \tau_s$ **then**

 4:  upload $\mathbf{x}$ to cloud server
                  ▷ *Upload challenging samples to Cloud*

 5: **end if**

 6: $\mathbf{y}_{\text{coarse}} \leftarrow f_{\text{up}}(\mathbf{y}_{\text{coarse}})$

 7: Estimate pixel-level uncertainty: $\mathbf{M} = \mathbf{Ent}(\mathbf{y}_{\text{coarse}})$
                     ▷ *Uncertainty decouple stage 2*

 8: **return** entropy uncertain mask $\mathbf{M}$, coarse prediction $\mathbf{y}_{\text{coarse}}$

  — **Collaborative Fusion Engine** —

 9: Refine the contour detatils: $\mathbf{y}_{\text{aug-coarse}} = \mathcal{M}_u(\mathbf{y}_{\text{coarse}}, \mathbf{M})$

10: **if** sample was processed on cloud **then**

11:  Receive fine-grained segmentation $\mathbf{y}_{\text{fine}}$ from cloud-side
                   ▷ *Receive output from Cloud*

12:  $\mathbf{y}_{\text{final}} \leftarrow \text{Fusion}(\mathbf{y}_{\text{aug-coarse}}, \mathbf{y}_{\text{fine}})$
                    ▷ *Dual-granularity fusion*

13: **else**

14:  $\mathbf{y}_{\text{final}} \leftarrow \mathbf{y}_{\text{aug-coarse}}$     ▷ Cloud processing skipped, use augmented coarse result

15: **end if**

16: **return** final output $\mathbf{y}_{\text{final}}$

---

**Algorithm 2** Segmentation Process on Cloud Server

---

**Input:** challenging sample $\mathbf{x}_t$, cloud model $\mathcal{M}_c$, uncertainty matrix $\mathcal{U}$

**Output:** fine-grained segmentatin $\mathbf{y}_{\text{fine}}$, updated uncertain matrix $\mathcal{U}_{\text{new}}$

  — **Dynamic Cloud Augmentation Module** —

 1: Receive challenging sample $\mathbf{x}_t$ from device-side
                   ▷ *Receive samples from Device*

 2: Calculate fine-grained segmentation: $\mathbf{y}_{\text{fine}} = \mathcal{M}_c(\mathbf{x}_t)$

 3: Get incorrectly classified classes $\mathbf{M}_c$ using $\mathbf{y}_{\text{fine}}$ as pseudo labels

 4: Calculate new uncertainty matrix $\mathcal{U}'$

 5: Update uncertainy matrix:
  $\mathcal{U}' \leftarrow \big((1-\alpha)\cdot\mathcal{U} + \alpha\cdot\mathcal{U}_{\text{new}}\big)\cdot\mathbf{M}_c + \mathcal{U}\cdot(1-\mathbf{M}_c)$
               ▷ *Adaptively updation of uncertainty decoupler*

 6: **return** fine-grained output $\mathbf{y}_{\text{fine}}$, updated uncertainty matrix $\mathcal{U}_{\text{new}}$
                 ▷ *Deliver the output $\mathbf{y}_{fine}$ to the device*

---

Algorithm 1 illustrates the device-side workflow, which comprises the DAS and CFE modules. DAS performs a coarse segmentation step and identifies challenging samples to upload to the cloud. CFE then refines contour details on the coarse predictions; if fine-grained predictions are received from the cloud, CFE fuses these dual-granularity outputs, otherwise it directly outputs the enhanced device-side predictions. Algorithm 2 presents the DCAM workflow on the cloud. DCAM processes the challenging samples for fine-grained segmentation and updates DAS's `uncertainty matrix` to adapt to environmental changes. Finally, the fine-grained predictions are sent back to the device for the final fusion step.

## A.2 Training and Inference Procedure

**Training Procedure.** The cloud model is designed exclusively for handling challenging samples and performing adaptive updates during inference; therefore, it does not participate in the training process. During training, we focus on optimizing the Uncertainty Matrix $\mathcal{U}$ and the Collaborative Fusion Engine (CFE) module using off-the-shelf segmentation models. The goal is to capture the probability distributions and extract contour details effectively.

For data preparation, input images are downsampled to half of their original resolution. The optimization objective is to minimize the loss function $\mathcal{L}_{\text{total}} = \mathcal{L}_{\mathcal{U}} + \mathcal{L}_{\text{U-Net}}$, where $\mathcal{L}_{\mathcal{U}}$ corresponds to the Uncertainty Matrix and $\mathcal{L}_{\text{U-Net}}$ corresponds to contour refinement. Stochastic Gradient Descent (SGD) is used as the optimizer, with an initial learning rate of 0.01 and a weight decay of 0.0005.

**Inference Procedure.** During inference, the cloud model collaborates with the device model to enhance segmentation performance. Input samples are initially downsampled on the device to generate coarse predictions. These predictions are processed through the Uncertainty Decoupler in the DAS module to separate model capability uncertainty from downsampling-induced uncertainty. Challenging samples identified by the Uncertainty Matrix are uploaded to the cloud's DCAM module for fine-grained segmentation. The DCAM module also updates the Uncertainty Matrix based on the refined cloud outputs. Simultaneously, the regions affected by downsampling-induced uncertainty are marked for contour information extraction. In the final step, the device model refines contour details based on the marked regions and combines these refined outputs with the cloud outputs using the proposed dual-granularity fusion mechanism in the CFE module. This fusion process produces the final accurate and efficient segmentation results.

## A.3 In-Depth Analysis

### A.3.1 Analysis between our method and baselines

We have previously discussed the performance of our method relative to baselines in Table 1. In this section, we delve deeper into the differences between them. For *Tent*Wang et al. (2020), a device-only method that performs test-time adaptive updates on normalization layers, it shows the poorest performance on most datasets except Pascal VOC. This result indicates that in segmentation tasks, where every pixel contributes to the final mIoU, direct parameter updates can severely disrupt previously learned knowledge, leading to a decline in performance. *CoTTA*Wang et al. (2022), a collaboration method refined from *Tent*, performs better by employing an Augmentation-Averaged Pseudo-Labels mechanism for guidance. However, as the number of augmentations increases, the latency also grows significantly, making it less suitable for efficient cloud-device collaboration.

For *Pseudo-label*Lee et al. (2013) and *CEMA*Chen et al. (2024), which are general-purpose collaboration methods, both outperform the *Base* model but yield only marginal gains. This shortfall stems from their reliance on entropy-based, image-level uncertainty detection, which fails to account for the pixel-wise granularity intrinsic to segmentation. *CDCCA*Wang et al. (2024), another general-purpose approach, employs a more sophisticated mechanism to identify challenging samples and achieves the best results among the baselines on ADE20K and Pascal VOC. However, it does not incorporate pixel-level feature information when estimating uncertainty, so many filtered samples remain within the device model's capability—thus limiting further performance improvements. *AMS*Khani et al. (2021), a segmentation-specific collaboration strategy, attains the highest baseline performance on Pascal Context by using a fixed-interval knowledge-distillation procedure tailored for segmentation. Yet, since it lacks a mechanism for detecting genuinely challenging samples, its fixed-interval updates can introduce substantial latency without delivering corresponding accuracy gains.

Our proposed *ACDC* employs a segmentation-specific uncertainty detection mechanism that considers the pixel-wise nature of segmentation tasks to detect distributional shifts in probability distributions. This mechanism enables the selective filtering of challenging samples and allows latency to be controlled through the filtration rate. Our method achieves state-of-the-art performance compared to the baselines. For datasets with a larger number of categories, such as ADE20K (150 categories) and Pascal Context (60 categories), segmentation tasks are more challenging and harder for device models to handle. Consequently, increasing the filtration rate from 5% to 10% results in substantial

improvements, with ACDC achieving gains of 2.94% and 5.01% in ADE20K and Pascal Context, respectively, compared to 2.27% and 2.71% in Cityscapes and Pascal VOC.

### A.3.2 IMPACT OF INCREASING FILTRATION RATE

Previously, we examined the effect of adaptive updates in DCAM by conducting experiments with varying filtration rates, highlighting the relationship between performance gains and the number of uploaded samples (see Figure 4). In this section, we delve deeper into how increasing filtration rates influence segmentation accuracy. As shown in Figure 5, for the Cityscapes and Pascal VOC datasets—which have relatively few classes—the device-side model can already handle most samples. Consequently, increasing the filtration rate from 0% to 10% yields only modest improvements, whereas raising it further to 20% produces more noticeable gains. In contrast, for datasets with many classes such as ADE20K and Pascal Context—which include a larger proportion of challenging samples beyond the device model's capabilities—a 10% filtration rate delivers substantial segmentation improvements. However, as the filtration rate continues to increase beyond 10%, the marginal gains diminish. This plateau occurs because higher filtration rates begin to admit samples that the device model can already process effectively, offering little to no additional benefit. These findings indicate that selecting an intermediate filtration rate is essential for balancing mIoU improvements against computational overhead. For practical applications, this balance ensures optimal performance without incurring unnecessary resource costs. Therefore, in our experiments, we adopt a 10% filtration rate uniformly across all datasets.

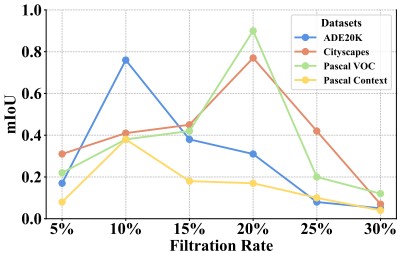

Figure 4: Gain on mIoU with increasing filtration rate.

### A.3.3 EFFECTS OF DIFFERENT MODULE COMBINATIONS

To evaluate the effects of the collaboration among all modules, we conducted an ablation study on various module combinations, as shown in Table 6. It is worth noting that the *DCAM+CFE* combination mimics a cloud-only model. While this configuration yielded strong performance, its high computational cost was contrary to our design goals for efficient processing. Therefore, we excluded it from further evaluation. The results indicate that our full ACDC configuration achieves the highest performance. The next best variant is *DAS+DCAM*. Compared to the base model, *DAS+CFE* also yields a noticeable improvement. The *DAS+DCAM* combination—where coarse- and fine-grained predictions are simply concatenated without additional refinement or fusion—approaches the cloud model's accuracy, resulting in a significant gain. In contrast, *DAS+CFE* operates solely on the device side, refining contour features based on the coarse predictions. This combination enhances coarse outputs by improving boundary delineation but lacks the fine-grained semantic detail provided by cloud processing. Consequently, although it outperforms the base, its performance trails that of the other two variants. By contrast, our complete ACDC framework synergistically integrates all three modules—DAS, DCAM, and CFE—merging fine-grained cloud predictions with refined coarse predictions to deliver the best overall segmentation accuracy. This ablation study thus confirms the effectiveness of our collaborative design.

### A.3.4 EFFECTS OF DIFFERENT DISTANCE METRICS

We propose the `Uncertainty Matrix` to model the distribution of model capabilities and employ its element-wise product with pixel-wise predictions to quantify distributional shifts. To evaluate our approach, we conducted ablation experiments comparing various distance metrics, as detailed in Table 7. Specifically, we benchmarked cosine similarity and Mahalanobis distance against

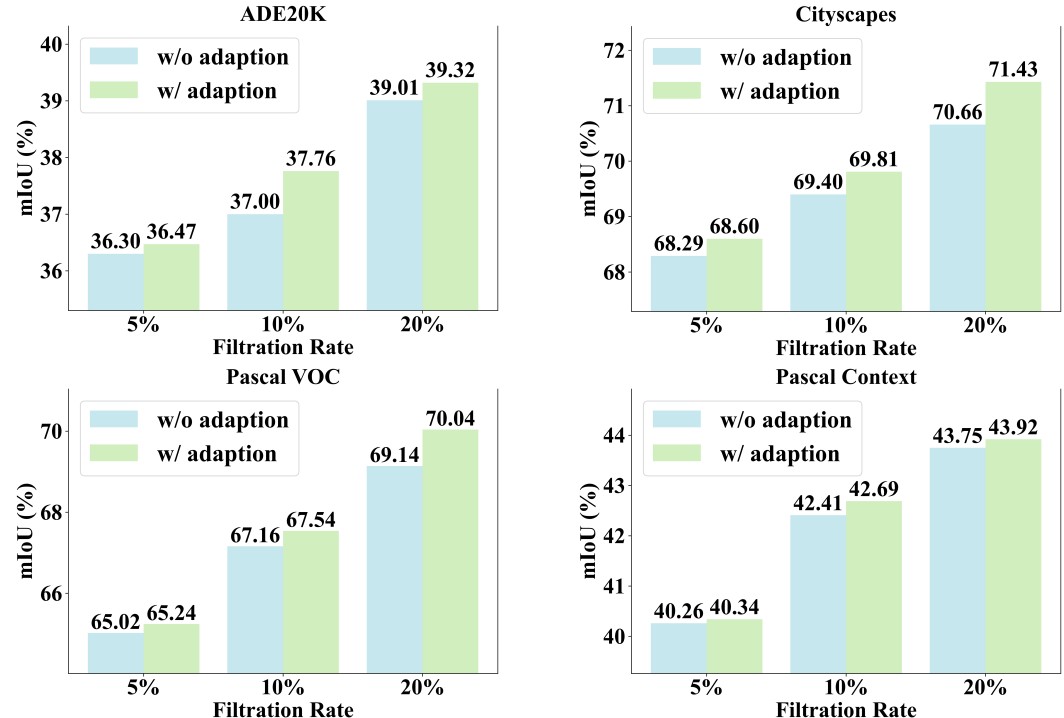

Figure 5: Performance of the adaptive update mechanism on the four datasets with varying filtration rates. "w/ adaptation" indicates that the parameters of the uncertainty matrix are updated using the adaptive mechanism, whereas "w/o adaptation" denotes static parameters without updates.

| Module | Datasets | | | |
| Combinations | ADE20K | Cityscapes | Pascal VOC | Pascal Context |
|---|---|---|---|---|
| Base | 35.25 | 65.30 | 63.47 | 39.34 |
| DAS+DCAM | 36.13 | 68.35 | 67.19 | 42.11 |
| DAS+CFE | 35.82 | 67.79 | 64.20 | 41.15 |
| **Ours** | **37.76** | **69.81** | **67.54** | **42.79** |

Table 6: Ablation study on the effects of different module combinations.

our method. The results demonstrate that, while these conventional metrics can partially capture shifts in parameter distributions, our proposed metric more effectively exploits pixel-level features to identify challenging samples in segmentation tasks, yielding superior performance overall.

| Methods | Datasets | | | |
| | ADE20K | Cityscapes | Pascal VOC | Pascal Context |
|---|---|---|---|---|
| Cosine | 37.14 | 69.61 | 66.88 | 41.79 |
| Mahalanobis | 37.59 | 69.63 | 67.10 | 42.13 |
| **Ours** | **37.76** | **69.81** | **67.54** | **42.78** |

Table 7: Ablation study on the effects of different distance metrics.

A.4 PROOF

A.4.1 PROOF OF UNCERTAINTY MATRIX

In this section, we provide a rigorous proof of the correctness of the loss function $\mathcal{L}(\mathcal{U}; \lambda, \gamma)$ and the reasonableness of the uncertainty matrix $\mathcal{U}$ in the context of sample-level uncertainty detection for segmentation tasks. The uncertainty score for an input $\mathbf{x}$ is defined as:

$$u(\mathbf{x}; \mathcal{U}) = \text{tr}\left(\mathbf{q}(\mathbf{x})\mathcal{U}\mathbf{q}(\mathbf{x})^\top\right), \tag{16}$$

where $\mathbf{q}(\mathbf{x}) \in \mathbb{R}^C$ is the class probability vector for the input $\mathbf{x}$, and $\mathcal{U} \in \mathbb{R}^{C \times C}$ is the uncertainty matrix.

**Properties of the Uncertainty Score** We first explore the properties of the uncertainty score $u(\mathbf{x}; \mathcal{U})$:

- **Linearity in $\mathcal{U}$:** The score is linear in $\mathcal{U}$, which means that the loss function is differentiable with respect to $\mathcal{U}$.

- **Trace Property:** The trace of a matrix is invariant under cyclic permutations, i.e., $\text{tr}(ABC) = \text{tr}(BCA)$.

**Loss Function Analysis** The loss function is defined as:

$$\mathcal{L}(\mathcal{U}; \lambda, \gamma) = (1 - \lambda) \cdot \mathbb{E}_{\mathbf{x}_+ \sim \mathcal{D}_+}\left[\mathcal{S}_\mathcal{U}(\mathbf{x}_+)\right] - \lambda \cdot \mathbb{E}_{\mathbf{x}_- \sim \mathcal{D}_-}\left[\mathcal{S}_\mathcal{U}(\mathbf{x}_-)\right] + \gamma\Omega(\mathcal{U}), \tag{17}$$

where $\lambda$ is the weighted factor, $\gamma > 0$ is the regularization strength, and $\Omega(\mathcal{U})$ is a regularization term.

**Objective of the Loss Function:** The objective of the loss function is to:

- Minimize the uncertainty score for positive samples ($\mathcal{D}_+$) to ensure confident predictions.

- Maximize the uncertainty score for negative samples ($\mathcal{D}_-$) to identify uncertain predictions.

This is achieved by:

- The first term $(1 - \lambda) \cdot \mathbb{E}_{\mathbf{x}_+ \sim \mathcal{D}_+}\left[\mathcal{S}_\mathcal{U}(\mathbf{x}_+)\right]$ encourages low uncertainty scores for positive samples.

- The second term $-\lambda \cdot \mathbb{E}_{\mathbf{x}_- \sim \mathcal{D}_-}\left[\mathcal{S}_\mathcal{U}(\mathbf{x}_-)\right]$ encourages high uncertainty scores for negative samples.

- The regularization term $\gamma\Omega(\mathcal{U})$ prevents overfitting by keeping the matrix $\mathcal{U}$ from becoming too large or complex.

**Optimization of the Loss Function:** To minimize the loss function, we take the derivative of $\mathcal{L}(\mathcal{U}; \lambda, \gamma)$ with respect to $\mathcal{U}$ and set it to zero:

$$\frac{\partial \mathcal{L}}{\partial \mathcal{U}} = (1 - \lambda) \cdot \mathbb{E}_{\mathbf{x}_+ \sim \mathcal{D}_+}\left[\mathbf{q}(\mathbf{x}_+)\mathbf{q}(\mathbf{x}_+)^\top\right] - \lambda \cdot \mathbb{E}_{\mathbf{x}_- \sim \mathcal{D}_-}\left[\mathbf{q}(\mathbf{x}_-)\mathbf{q}(\mathbf{x}_-)^\top\right] + 2\gamma\mathcal{U} = 0. \tag{18}$$

Solving for $\mathcal{U}$, we get:

$$\mathcal{U} = \frac{(1 - \lambda) \cdot \mathbb{E}_{\mathbf{x}_+ \sim \mathcal{D}_+}\left[\mathbf{q}(\mathbf{x}_+)\mathbf{q}(\mathbf{x}_+)^\top\right] - \lambda \cdot \mathbb{E}_{\mathbf{x}_- \sim \mathcal{D}_-}\left[\mathbf{q}(\mathbf{x}_-)\mathbf{q}(\mathbf{x}_-)^\top\right]}{2\gamma}. \tag{19}$$

**Interpretation of the Uncertainty Matrix** The uncertainty matrix $\mathcal{U}$ captures the relationships between different classes based on the uncertainty of the model's predictions. Specifically:

- **Positive Definiteness:** If $\mathcal{U}$ is positive definite, it ensures that the uncertainty score is always non-negative, which is desirable for uncertainty measurement.

- **Symmetry:** The matrix $\mathcal{U}$ is symmetric, which is a natural property for measuring distances or uncertainties between classes.

By minimizing the loss function $\mathcal{L}(\mathcal{U}; \lambda, \gamma)$, we effectively learn an uncertainty matrix $\mathcal{U}$ that distinguishes between positive and negative samples by assigning lower uncertainty scores to positive samples and higher uncertainty scores to negative samples. This ensures that the model can effectively identify and filter out samples with high uncertainty, thereby improving the overall performance of the segmentation task.

### A.4.2 GENERALIZATION ERROR UPPER BOUND FOR THE PROPOSED MODEL

In this section, we derive the generalization error upper bound for the proposed cloud-device collaborative semantic segmentation model. We consider the device model $M_{\text{device}}$ and the cloud model $M_{\text{cloud}}$, where only $M_{\text{device}}$ is trained on the training set, while both models collaborate during inference on the test set.

**Definitions and Assumptions**

- **Empirical Error**: The empirical error on the training set is defined as:

$$\hat{R}_{\text{train}}(M_{\text{device}}) = \frac{1}{n} \sum_{i=1}^{n} \mathcal{L}_{\text{total}}(M_{\text{device}}; \mathbf{x}_i, \mathbf{y}_i), \tag{20}$$

where $\mathcal{L}_{\text{total}} = \mathcal{L}_{\mathcal{U}} + \mathcal{L}_{\text{U-Net}}$ is the total loss function, and $(\mathbf{x}_i, \mathbf{y}_i)$ are training samples.

- **Generalization Error**: The generalization error is defined as the expected loss on the test set:

$$R_{\text{test}}(M_{\text{device}}, M_{\text{cloud}}) = \mathbb{E}_{\mathbf{x},\mathbf{y} \sim \mathcal{D}} \left[ \mathcal{L}_{\text{total}}(M_{\text{device}}, M_{\text{cloud}}; \mathbf{x}, \mathbf{y}) \right], \tag{21}$$

where $\mathcal{D}$ is the data distribution.

**Assumption 1.** *The training samples are independently and identically distributed (i.i.d.) from the distribution $\mathcal{D}$.*

**Assumption 2.** *The loss function $\mathcal{L}_{total}$ is bounded, i.e., $0 \leq \mathcal{L}_{total} \leq B$ for some constant $B$.*

**Hoeffding's Inequality and Joint Probability Bound**   To derive the generalization error upper bound, we use Hoeffding's inequality, which provides a bound on the deviation of the empirical mean from the true mean.

**Theorem 1** (Hoeffding's Inequality). *For i.i.d. samples $Z_1, Z_2, \ldots, Z_n$ with $\mathbb{E}[Z_i] = \mu$ and $a_i \leq Z_i \leq b_i$, we have:*

$$\Pr\left( \left| \frac{1}{n} \sum_{i=1}^{n} Z_i - \mu \right| \geq \epsilon \right) \leq 2 \exp\left( -\frac{2n^2\epsilon^2}{\sum_{i=1}^{n}(b_i - a_i)^2} \right). \tag{22}$$

In our case, since $0 \leq \mathcal{L}_{\text{total}} \leq B$, we can set $a_i = 0$ and $b_i = B$ for all $i$.

**Derivation of the Generalization Error Upper Bound**   We aim to bound the difference between the empirical error and the generalization error:

$$\mathcal{E}_{\text{gen}} = \left| \hat{R}_{\text{train}}(M_{\text{device}}) - R_{\text{test}}(M_{\text{device}}, M_{\text{cloud}}) \right|. \tag{23}$$

Using Hoeffding's inequality, we have:

$$\Pr\left( \left| \hat{R}_{\text{train}}(M_{\text{device}}) - R_{\text{test}}(M_{\text{device}}, M_{\text{cloud}}) \right| \geq \epsilon \right) \leq 2 \exp\left( -\frac{2n\epsilon^2}{B^2} \right). \tag{24}$$

To account for the collaboration between $M_{\text{device}}$ and $M_{\text{cloud}}$ during inference, we consider the joint probability of the errors from both models. Let $\epsilon_{\text{device}}$ and $\epsilon_{\text{cloud}}$ be the generalization errors for $M_{\text{device}}$ and $M_{\text{cloud}}$, respectively. Using the union bound, we have:

$$\Pr\left( \epsilon_{\text{device}} \geq \epsilon \text{ or } \epsilon_{\text{cloud}} \geq \epsilon \right) \leq \Pr\left( \epsilon_{\text{device}} \geq \epsilon \right) + \Pr\left( \epsilon_{\text{cloud}} \geq \epsilon \right). \tag{25}$$

Assuming both models have the same generalization error bound, we get:

$$\Pr\left(\epsilon_{\text{device}} \geq \epsilon \text{ or } \epsilon_{\text{cloud}} \geq \epsilon\right) \leq 4\exp\left(-\frac{2n\epsilon^2}{B^2}\right). \tag{26}$$

Thus, with high probability, the generalization error is bounded by:

$$R_{\text{test}}(M_{\text{device}}, M_{\text{cloud}}) \leq \hat{R}_{\text{train}}(M_{\text{device}}) + \epsilon, \tag{27}$$

where

$$\epsilon = \sqrt{\frac{B^2 \ln(4/\delta)}{2n}} \tag{28}$$

for some confidence level $1 - \delta$.

**Incorporating Uncertainty Matrix and CFE Module**  The uncertainty matrix $\mathcal{U}$ and the CFE module play crucial roles in identifying challenging samples and fusing the predictions, respectively. The dimensions and variables involved in these modules directly influence the generalization error. Specifically, the uncertainty matrix $\mathcal{U}$ has shape $C \times C$, where $C$ represents the number of classes. The CFE module involves the fusion of coarse and fine predictions, which introduces additional variables such as the fusion weights and the granularity of the fusion process.

By accurately identifying challenging samples through $\mathcal{U}$, the model reduces the generalization error by focusing on difficult regions during inference. The fusion process in the CFE module further refines the predictions, thereby improving the overall generalization performance.

The derived generalization error upper bound provides insights into the impact of the uncertainty matrix and the CFE module on the model's performance. The bound shows that the error decreases as the number of training samples $n$ increases, and it depends on the confidence level $\delta$ and the bound $B$ on the loss function. This analysis highlights the importance of the proposed cloud-device collaborative approach in improving the generalization performance of semantic segmentation tasks.

## A.5  USE OF LARGE LANGUAGE MODELS (LLMs)

It is important to note that the LLM was not involved in the ideation, rescarch methodology, writing, or experimental design. All research concepts, ideas, and analyses were developed and conducted by the authors.

