# OpenReview forum: "ACDC: Adaptive Cloud-Device Collaboration for Efficient and Accurate Semantic Segmentation"
_ICLR.cc/2026/Conference — Submitted to ICLR 2026_

### Official Review · Reviewer_Yb8V · 2025-10-28

**Soundness:** 2
**Presentation:** 3
**Contribution:** 2
**Rating:** 4
**Confidence:** 4

**Summary:**

This paper proposes ACDC, a framework for cloud–device collaborative semantic segmentation that aims to balance accuracy and efficiency. The framework consists of three main components:

(1) A Device-Aware Adaptive Segmentor (DAS), which performs lightweight, coarse segmentation on devices and identifies uncertain samples through a two-stage Uncertainty Decoupler that distinguishes uncertainty due to model limitations and downsampling effects.
(2) A Dynamic Cloud Augmentation Module (DCAM), which processes the uncertain samples on the cloud, providing fine-grained segmentation and adaptive updates to the device’s uncertainty matrix.
(3) A Collaborative Fusion Engine (CFE), which merges coarse and fine-grained outputs through a dual-granularity fusion mechanism.

Experiments on ADE20K, Cityscapes, Pascal VOC, and Pascal Context show consistent mIoU improvements compared to cloud-only and prior collaborative baselines (e.g., CDCCA, AMS, CEMA), while reducing latency and FLOPs.

**Strengths:**

1. The ACDC framework integrates uncertainty estimation, cloud offloading, and multi-level fusion in a coherent architecture. Each module serves a clear functional role, and the overall system is well-motivated by practical deployment constraints.
2. The two-stage Uncertainty Decoupler introduces a nuanced separation between model and sampling uncertainties. This addresses a genuine gap in existing collaboration approaches, which often use entropy or confidence-based heuristics that are not well-suited to pixel-wise tasks.
3. The experiments are broad and well-organized, spanning multiple datasets and baselines. Quantitative results are supported by qualitative visualizations (Fig. 3). The framework demonstrates consistent accuracy improvements and measurable reductions in latency and communication load.
4. The paper is well-written, the motivation is easy to follow, and the figures effectively illustrate complex components (especially Fig. 2).

**Weaknesses:**

1. While the components (uncertainty estimation, adaptive update, fusion) are each reasonable, the paper primarily integrates established ideas under a unified framework rather than introducing a fundamentally new theoretical concept. The contribution is mostly system-level engineering rather than algorithmic innovation.

2. The proposed “uncertainty matrix” formulation uses contrastive objectives and Frobenius norms, but the intuition behind Eq. (6–9) remains underexplained. It is unclear how well this formulation captures epistemic versus aleatoric uncertainty or how sensitive it is to hyperparameters such as λ, γ, or the kernel width. A more rigorous or probabilistic interpretation would strengthen the contribution.

3. Although the method claims minimal overhead, the paper provides limited quantitative evidence about the communication cost, training time, or memory footprint. Since efficiency is a central claim, a clear runtime–accuracy curve or ablation on upload bandwidth would make the results more convincing.

4. All experiments are conducted on canonical segmentation datasets using a fixed HRNet–SegFormer pair. There is no evaluation under real-world dynamic or continual settings, despite the stated goal of adapting to “dynamic environments.” This weakens the paper’s central claim of adaptivity and practical relevance.

5. The paper reports quantitative gains but provides little qualitative analysis of the uncertainty matrix or the behavior of the filtration rate.
For example, visualizing where and why the decoupler flags uncertainty would help validate its conceptual soundness.

6. Competing uncertainty-based or probabilistic collaboration frameworks (e.g., Bayesian or ensemble-based segmentation) are not included in the comparison. Without these, it is difficult to attribute the gains specifically to the proposed uncertainty decoupling mechanism.

7. The choice of HRNet (device) and SegFormer (cloud) seems tuned for complementary behavior. It is unclear whether ACDC’s performance persists if both sides use similar architectures or if the cloud model is only moderately stronger.


Overall, the paper tackles a practically important problem—efficient semantic segmentation across device–cloud boundaries—and offers a technically solid, well-tested framework. The system is coherent and reproducible, and the performance gains are empirically verifiable. However, the theoretical underpinnings of the uncertainty modeling remain somewhat shallow, and the adaptivity claims are stronger than what the experiments actually demonstrate. In essence, this is a high-quality system paper with limited conceptual novelty but clear empirical value.

**Questions:**

See weeknesses

---

### Official Review · Reviewer_mwcQ · 2025-10-29

**Soundness:** 3
**Presentation:** 1
**Contribution:** 2
**Rating:** 2
**Confidence:** 3

**Summary:**

Due to the significant disparity in computational resources between cloud and device, the authors propose ACDC to address the challenges of semantic segmentation in cloud-device collaboration, ensuring high segmentation performance while enabling efficient inference. The core concept is to enable device-based models to distinguish between easy and difficult samples, processing simple images rapidly, while the cloud-based model handles complex ones and returns results. Ultimately, the final fusion segmentation outcome is achieved through the transmission of edge information.

**Strengths:**

The proposed method has been validated across multiple datasets.

**Weaknesses:**

1. The key limitations and key challenges in the introduction convey nearly identical meanings.

2. The format of citations is wrong without spacing and parentheses.

3. Excessive font formatting in the introduction, such as **bold**, *italic*, $\underline{underline}$, and $\texttt{texttt}$.

4. No corresponding explanation follows the appearance of Eq. (3).、

5. In Section 3.2, the statement “we propose the Device-Aware Adaptive Segmentor (DAS) and the Collaborative Fusion Engine (CFE) to achieve efficient, high-quality segmentation” is ambiguous. “Achieve efficient, high-quality segmentation” refers to the CFE.

6. In Eq. (4), $s$ should be treated as the input to the $f_{down}$ function. What is the upsample function for $\text{y}_\text{coarse}$?

7. Why is the last dimension of Eq. (4) denoted as $C$? Shouldn't $C$ represent the input dimension? Additionally, why is the uncertainty matrix also defined as $C \times C$?

8. What distinguishes $\text{y}_\text{coarse}$ from $p(x)$?

9. What do $y$ and $\hat{y}$ represent, respectively? How do they differ from $\text{y}_\text{coarse}$?

**Questions:**

Due to the numerous ambiguities and unclear points in the writing, I am unable to understand the author's approach accurately. Please see above.

---

### Official Review · Reviewer_Jsem · 2025-10-30

**Soundness:** 3
**Presentation:** 2
**Contribution:** 2
**Rating:** 4
**Confidence:** 3

**Summary:**

This manuscript introduces ACDC (Adaptive Cloud-Device Collaboration), a novel framework for efficient and accurate semantic segmentation that integrates device-side and cloud-side processing. The key idea is to leverage adaptive uncertainty detection to decide which samples should be offloaded to the cloud for fine-grained refinement.  Experiments on several semantic segmentation datasets demonstrate consistent improvements in mIoU, reduced latency, and minimized upload bandwidth compared to both device-only and prior cloud-device collaboration baselines. The manuscript also provides thorough ablations and theoretical grounding for the Uncertainty Matrix.

**Strengths:**

1. Rationality: The methodology is conceptually sound and well-motivated. The uncertainty modeling and adaptive updates are mathematically justified (with a detailed proof in the appendix).
2. Novelty: The Uncertainty Matrix effectively models pixel-wise distributional shifts, outperforming prior entropy-based approaches. The DCAM’s weighted update mechanism allows continual adaptation under distribution shifts.
3. Comprehensive evaluation: Demonstrated improvements across four major datasets with consistent gains in mIoU, latency, and FLOPs. Extensive analysis of filtration rate, module combinations, and distance metrics solidifies the claims that the three modules can be integrated with various segmentation backbones.

**Weaknesses:**

1. Complex training pipeline: The optimization of the Uncertainty Matrix and adaptive updates might hinder reproducibility and real-time deployment feasibility.
2. Theoretical depth: While the Uncertainty Matrix is intuitive, its probabilistic grounding could be connected more clearly to existing Bayesian or information-theoretic frameworks.
3. Scope limitation: Focused solely on semantic segmentation; applicability to other pixel-level or multimodal tasks (e.g., depth estimation, panoptic segmentation) is not discussed. Comparisons with recent sparse refinement methods (e.g., SparseRefine) are missing.

**Questions:**

1．How sensitive is the performance to the choice of filtration threshold (τₛ)? Can this threshold be learned dynamically rather than tuned manually?
2. Could the Uncertainty Matrix be approximated or compressed to reduce its computational footprint on-device?
3. Is there a feedback delay consideration in DCAM updates, and how does it affect online adaptation?
4. How would ACDC perform under severe bandwidth constraints or intermittent connectivity?
5. Could this framework be extended to multimodal segmentation (e.g., RGB + depth)?
6. Why choose a lightweight convolutional layer in CFE instead of attention mechanisms for fusion?

---

### Official Review · Reviewer_yiy7 · 2025-11-03

**Soundness:** 3
**Presentation:** 2
**Contribution:** 2
**Rating:** 2
**Confidence:** 4

**Summary:**

This paper proposes ACDC, a cloud-device collaboration framework for semantic segmentation that balances on-device efficiency with cloud-based accuracy. It uses a lightweight device model for fast, coarse predictions and a heavyweight cloud model for high-accuracy processing.
The core of ACDC is a two-stage uncertainty decoupler that identifies samples from shifted distributions or in general more challenging ones and filters what to send to the cloud:

(1) sample-level uncertainty: a learnable uncertainty matrix is proposed for identifying challenging samples (due to the limited capacity of the model ) and offloads them entirely.

(2) pixel-level uncertainty: a standard per-pixel entropy check identifies uncertain pixels (due to downsampling) for local, on-device refinement.

The ACDC framework is completed by a Dynamic Cloud Augmentation Module (DCAM), which adaptively updates the device's uncertainty matrix to refine it on wrong predicted classes (as estimated by difference in pseudo-labels) and sends it back to the local device, and a Collaborative Fusion Engine (CFE), which efficiently fuses the coarse device mask with the fine-grained cloud mask.

Experiments on 4 semantic segmentation datasets (ADE20K, Cityscapes, Pascal VOC, Pascal Context) show ACDC outperforms baselines in mIoU while minimizing data transmission.

**Strengths:**

**Significance**
- This work addresses a practical and important problem: enabling high-accuracy segmentation on resource-constrained devices. The cloud-device collaboration paradigm is a pragmatic and efficient solution for real-world applications where latency and local computational resources are critical bottlenecks.


**Originality**
- The Uncertainty Decoupler that separates uncertainty into "model capability" (sample-level) vs. "information loss" (pixel-level) seems novel to me.
- The uncertainty matrix to estimate per image difficulty for a semantic segmentation model is interesting and novel

**Quality**
- The framework is validated on four standard benchmarks (ADE20K, Cityscapes, Pascal VOC, Pascal Context).
- The authors report performance, latency and FLOPs gains, that give an interesting trade-off in the 10% filtration rate

**Weaknesses:**

**Unclear core contribution**
- I find that the paper's core contribution, the uncertainty matrix and its associated loss function (Eq. 4-9), is mathematically dense and poorly motivated
- The paper does not provide a clear, intuitive derivation for why the score function $\mathcal{S}_{U}(x)$ (Eq. 7) or the smoothing function $\mathcal{Q}(z)$ (Eq. 8) are correct formulations for this problem. They are presented without justification, making it difficult to understand what the learned matrix U is actually modeling. There is zoom-in on their form in Appendix A.4.1 but essentially details the equations and puts them together
- For the final uncertainty score $u(x;\mathcal{U}) = tr(q(x)\mathcal{U}q(x)^{\top})$ (Eq. 9) the paper just states it "reflects the uncertainty" without explaining why this specific metric is superior to simpler, well-understood metrics like Mahalanobis distance, which is also designed to measure distributional shifts. The ablation in Table 7 is insufficient as it doesn't explain why the proposed metric works better and the gaps with Mahalanobis are small.

**Limited related work**
- When reading the description of the uncertainty decoupler and how it aims to separate uncertainty from model limitation and information loss/ambiguity from downsampling one can easily get the feeling of "reinventing the wheel" from uncertainty estimation literature [a], [b], [c]. We already have a large body of works in formalizing and naming the types of uncertainty (epistemic/knowledge, aleatoric/data) and it's not clear why the paper prefers going on its own way with the definitions and the formalism.


**Limited evaluation**
- the paper mentions potential OOD images, that lie outside the training distribution, when motivating their method. However all experiments are conducted in in-distribution mode. The utility of uncertainty is more obvious in conditions of distribution shift, in particular on real-world applications such as this one proposed here.
- I would recommend looking at different forms of distribution shift and there are lots of  ready to use variants compatible with Cityscapes that can be used: Cityscapes-C (with image corruptions) [d],[e], ACDC [f]  (with different weather conditions: snow, rain, night, fog), BRAVO [g] (with different shifts of distribution and perturbations).
- Given that the method assumes that a large model can run on the cloud, it would be interesting to explore heavier backbones or uncertainty estimation methods on that side, in particular since this can improve the performance locally, by updating the uncertainty matrix. Currently only one backbone is considered for local and cloud.
- Also it would be interesting to know what is the upper bound for this setting and how far/close is the current method from that. To this end the results of the cloud-only variant could be reported.


**Small performance gains**
- The proposed uncertainty decoupler module to select uncertain samples shows modest improvements compared to the simple entropy baseline (Table 2), between +0.15 (on Pascal) and +0.64 at best (on Cityscapes). As the evaluation does not containt shifted distribution samples it's difficult to say whether this is due to the limited difficulty of the data or the limited efficacy of the proposed module
- Similarly, the adaptive update mechanism shows limited performance gains, in particular on the 5% filtration rate ((+0.08 to +0.17), where one would expect that the most difficult samples to be dealt with and the biggest boosts. On 10% and 20% the relative improvements area bit better.


**Clarity**
- It's not clear why the trace (Eq. 9) reflects the uncertainty
- It's not clear how the baselines in Table 2 entropy and CEMA were computed, as they were originally proposed for image classification.


**[Minor] Related work**
- The use of class prototypes for estimation of distribution shift is slightly related with some methods on deterministic uncertainty estimation [h], [i]


**[Minor] Misc.**
- With natbib please use $\citep$ accordingly for the references
- The use of $c$ subscript is confusing as it's used for "cloud", "coarse"
- K and C are both used for the number of classes
- In Figure 1 the dataset on which the scores were computed is not mentioned
- The layout and organization of the paper could be improved: the tables with results arrive way ahead of the experiments section
- Figure 3 would need a column with the ground truth labels
- The ACDC names of the method clashes with the dataset with the same name that is quite established in this area [f]


**References:**

[a] Kendall & Gal, What Uncertainties Do We Need in Bayesian Deep Learning for Computer Vision?, NeurIPS 2017

[b] Hullermeier & Waegeman, Aleatoric and epistemic uncertainty in machine learning: An introduction to concepts and methods, ML 2021

[c] Seng et al., Reliable classification: Learning classifiers that distinguish aleatoric and epistemic uncertainty, IS 2014

[d] Hendrycks & Dietterich, Benchmarking Neural Network Robustness to Common Corruptions and Perturbations, ICLR 2019

[e] Franchi et al., Robust Semantic Segmentation with Superpixel-Mix, BMVC 2021

[f] Sakaridis et al., ACDC: The Adverse Conditions Dataset with Correspondences for Robust Semantic Driving Scene Perception, ICCV 2021

[g] Vu et al., The BRAVO Semantic Segmentation Challenge Results in UNCV2024, ECCV Workshops 2024


[h] Macedo et al., Entropic Out-of-Distribution Detection, ICNN 2021

[i] Franchi et al., Latent Discriminant deterministic Uncertainty, ECCV 2022

**Questions:**

This paper takes an interesting direction of study: a cloud-device collaboration framework for semantic segmentation that balances on-device efficiency with cloud-based accuracy. The idea itself make sense and the authors identify multiple of the good ingredients for such a solution: uncertainty estimation and decoupling, cloud-level prediction and refinement, local update and fusion of predictions. However I do have several concerns regarding the limited experiments, essentially all conducted in-distribution, and the limited performance gains, while using a fairly complex approach.

My current rating is leaning towards reject at this time, but I'm looking forward for the rebuttal.

Here are a few questions and suggestions that could be potentially addressed in the rebuttal or in future versions of this work (please note that suggested experiments are not necessarily expected to be conducted for the rebuttal):

1. Consider adding some results with data actually shifted from the train distribution, e.g., Cityscapes-C, ACDC, BRAVO subsets, etc.

2. Add discussion on why the trace is a good proxy for uncertainty and in general a discussion on the reasoning of the proposed uncertainty estimation strategy.

3. Add description of the implementation of the Entropy and CEMA baselines from Table 2.

4. Add discussion on the relationship between ACDC and different types of uncertainty (epistemic, aleatoric)

---

### Meta-Review · Area_Chair_maKQ · 2025-12-03

**Summary:**

The reviewers agree that the paper addresses a practically relevant problem, and that the overall system design is coherent and technically reasonable. However, they also raise several major concerns:

1. The main technical contribution is not clearly motivated or well explained. Key equations and metrics are introduced with limited intuition or probabilistic grounding, and the paper does not convincingly connect the approach to existing works, presenting limited conceptual novelty.

2. The experimental validation does not fully support the paper’s claims. Although the method is motivated by robustness to distribution shift and dynamic environments, the evaluation is restricted to standard in-distribution segmentation benchmarks, without tests under realistic shifts or dynamic/online settings. Stronger, more recent baselines are missing from comprehensive comparisons. In addition, core efficiency aspects such as communication cost, runtime, and memory overhead are not systematically measured or analyzed.

3. Reviewers also point out notable clarity and reproducibility issues, including ambiguous notation, missing explanations around key equations, organizational problems in the presentation, incomplete details on baseline implementations, etc. The authors did not submit a rebuttal, so these major concerns remain unaddressed. As a result, the AC recommends rejecting the paper in its current form.

**Reviewer Concerns:**

Since no rebuttal is provided, all the reviewers' concerns are still outstanding.

**Reviewer Scores:**

Reviewer yiy7: 2
Reviewer Jsem: 4
Reviewer mwcQ: 2
Reviewer Yb8V: 4

---

### Decision · Program_Chairs · 2026-01-26

Reject